# TRAINING NEURAL NETWORKS WITH LOW-PRECISION MODEL MEMORY

## ABSTRACT

The demand for memory to store model-related statistics ("model memory") is a major bottleneck for training large neural networks. A promising solution is low-precision optimizers, which reduce the numerical precision of the model memory. However, existing work only compresses the momentum, resulting in suboptimal memory efficiency. This paper proposes Low-Precision Model Memory (LPMM), an optimization framework with the entire model memory kept in low precision. LPMM compresses not only the momentum but also model parameters and gradient accumulators. We identify arithmetic underflow as the main problem in building low-precision optimizers and propose a stochastic quantization method and a microbatching technique to overcome this problem. We analyze the convergence behavior of LPMM and theoretically show how the proposed techniques could affect underflowing, which in turn affects the convergence. We apply LPMM to the SGD optimizer with momentum (SGDM). On several realistic benchmarks, LPMM-SGDM can train neural networks with negligible loss of accuracy while reducing over 70% of the model memory compared to the full-precision SGDM.

## 1 INTRODUCTION

Many huge models (Kenton & Toutanova, 2019; Radford et al., 2019; Dosovitskiy et al., 2021; Brown et al., 2020; Fedus et al., 2022) emerge in the recent several years. Despite being powerful, training these models is challenging. Memory is the main bottleneck in developing large models, as the training memory footprint is typically proportional to the number of model parameters.

During training, the device memory is consumed by three types of objects:

1. **data-related objects** ("data memory"), including data and each layer's activation. Their size is proportional to the data size, i.e., mini-batch size and image resolution / sequence length.
2. **model-related objects** ("model memory"), including model parameters, momentum, and gradient accumulators. Their size is proportional to the amount of model parameters.
3. **temporary objects** ("workspace memory"), such as scratch memory used by computing kernels and memory fragments.

Among the three types, model memory is the main bottleneck in scaling up machine learning models (Rajbhandari et al., 2020).

Quantization is a promising way of reducing the model memory. Specifically, *low-precision optimizers* (Ramesh et al., 2021; Dettmers et al., 2021) represent their states with low-precision numerical formats, such as 8-bit integers, which consume less memory. Particularly, Dettmers et al. (2021) propose an 8-bit optimizer, which quantizes the momentum to a block-wise 8-bit format. However, existing works have two limitations. First, the convergence behavior of low-precision optimizers is theoretically not well understood. Second, they only quantize the momentum, while model parameters and gradients are left in full precision. Therefore, the overall memory saving is unsatisfactory.

In this work, we propose LPMM, a novel framework for optimizing with *low-precision model memory*. Unlike previous works, LPMM consider all model-related objects, including model parameters, momentum, and gradients, in low precision. We identify *arithmetic underflow* as the major bottleneck of building low-precision optimizers. Theoretically, we analyze the convergence behavior of low-precision optimizers. Our analysis reveals how the design of the optimizer could impact the

degree of underflowing, which we link to the convergence behavior. Algorithmically, we propose stochastic quantization and gradient accumulation methods to reduce underflowing. These techniques are backed with our theoretical findings. We further discuss the quantizer design and system implementation for LPMM.

We evaluate LPMM on the standard image classification benchmark. LPMM can quantize gradients and the momentum to 8 bit, and model parameters to 12 bit, with neglible loss of accuracy. In total, LPMM only requires 28 bits of model memory per parameter, compared to 72 bits by Dettmers et al. (2021) or 96 bits of the full-precision algorithm.

## 2 RELATED WORK

**Training Methods With Compressed Model Memory**  Several works train neural networks with quantized parameters, momentum, or gradients. QSGD (Alistarh et al., 2017) quantizes the gradient into lower bits for efficient communication. Low-Precision SGD Li et al. (2017); Li & De Sa (2019); De Sa et al. (2018); Yang et al. (2019) uses low-precision parameters for training weight-quantized neural networks. The 8-bit Optimizer (Dettmers et al., 2021) quantizes the momentum for SGDM and Adam. Many of these works are not designed for saving the memory. Moreover, they only consider different parts of the model memory, without unified framework and analysis.

**Other Methods to Reduce the Model Memory**  Sharding (Rajbhandari et al., 2020) splits model parameters across multiple nodes, which is only applicable to distributed settings. Offloading (Ren et al., 2021; Rajbhandari et al., 2021) stores model-related statistics in CPU memory. These methods have heavy communication overhead, which can be reduced by our method.

**Other Memory Efficient Training Methods**  Gradient checkpointing (Chen et al., 2016) reduces the storage overhead of activations by recomputation. Activation Compressed Training (Chen et al., 2021; Liu et al., 2022) keeps low-precision activations in memory. These method reduces the data memory rather than the model memory, and are orthogonal to our approach.

**Quantized Training**  There is a series of work focusing on the acceleration of neural network training with low-precision computations (Wang et al., 2018b; Micikevicius et al., 2017; Zhu et al., 2020; Sun et al., 2020). However, they still store the model-related objects in full-precision or 16-bit.

## 3 TRAINING WITH LOW-PRECISION MODEL MEMORY

In this section, we formulate the problem of training neural networks with low precision memory. We show that *arithmetic underflow* is the major bottleneck for reducing the numerical precision. To solve this problem, we propose stochastic quantization and micro-batching techniques. Here, we mainly consider stochastic gradient descent with momentum (SGDM) (Qian, 1999; Sutskever et al., 2013) as a motivating example, but the proposed techniques also apply to other optimizers, such as stochastic gradient descent (Bottou, 2010) and Adam (Kingma & Ba, 2015).

### 3.1 BASIC OPTIMIZATION FRAMEWORK

Consider the empirical risk minimization problem

$$\min_{\theta} f(\theta) = \frac{1}{n} \sum_{i=1}^{n} f_i(\theta), \tag{1}$$

where $\theta$ is the model parameter. Since the dataset size $n$ is large, stochastic optimizers are adopted for solving the problem. SGDM is one of the most widely used optimizers. Starting with an initial model $\theta_0$ and momentum $m_0 = 0$, SGDM performs the following updates:

$$m_t \leftarrow \beta m_{t-1} + \nabla \tilde{f}(\theta_{t-1}), \quad \theta_t \leftarrow \theta_{t-1} - \alpha m_t,$$

where $\nabla \tilde{f}(\theta_{t-1})$ is an unbiased estimator of the gradient. We define $\nabla \tilde{f}(\theta_{t-1}) := \nabla f(\theta_{t-1}, \zeta^t) := \frac{1}{|\zeta^t|} \sum_{i \in \zeta^t} \nabla f_i(\theta_{t-1})$, where $\zeta^t$ is a minibatch sampled uniformly from the dataset $\{1, 2, \ldots, n\}$.

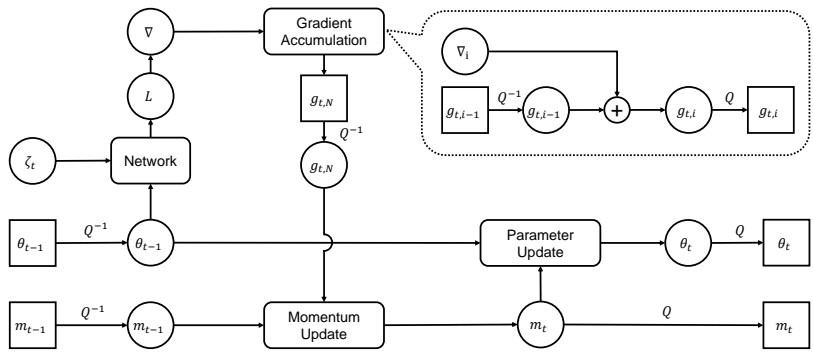

Figure 1: Dataflow of the LPMM algorithm. Square nodes are in low-precision and kept persistent in memory. Circular nodes are temporary full-precision data, which only consume a small constant-size (dimension-free) buffer.

When training large neural networks, the dimensionality of $\theta$ is huge. Storing the model-related statistics, including the model $\theta$, the momentum $m$, and the gradient $\nabla \tilde{f}(\theta)$ can be a major memory overhead. We call these statistics collectively as the *model memory*. For example, the GPT-2 (Radford et al., 2019) model has 1.5 billion parameters. During training, each parameter requires $32(\text{model}) + 32(\text{grad.}) + 32(\text{momentum})$ bits or 12 bytes of model memory. Therefore, the model memory alone takes about $1.5 \times 10^9 \times 12\text{bytes} = 17.6\text{GB}$ of GPU memory, which already exceeds the capacity of many mainstream GPUs.

To reduce training memory footprint, we can store the model-related statistics in low numerical precision. This involves a *quantizer* $Q(\cdot)$, which encodes full-precision (32-bit) values to lower-precision ones, such as 8-bit integers. The training procedure can be summarized as:

$$g_t \leftarrow Q\left(\nabla \tilde{f}(\theta_{t-1})\right), \quad m_t \leftarrow Q\left(\beta m_{t-1} + \nabla \tilde{f}(\theta_{t-1})\right), \quad \theta_t \leftarrow Q\left(\theta_{t-1} - \alpha m_t\right). \quad (2)$$

Here, $g_t$ is a gradient accumulator, which stores the temporary gradients during back-propagation. When the parameters and momentum are updated, they are first decoded to full-precision, computed in full-precision, and then encoded and stored in low-precision, as illustrated in Fig. 1. We defer further implementation details to Sec. 5.3.

### 3.2 THE UNDERFLOW PROBLEM

Quantization is lossy, which can only represent values up to a numerical precision. Consider the following symmetric linear quantizer $Q = Q_{\delta,B} : \mathbb{R} \to \mathbb{R}$ as an example:

$$Q_{\delta,B}(x) = \delta \cdot \text{Round}\left(\text{clip}\left(x/\delta, -B, B\right)\right), \quad (3)$$

where $\text{Round}(\cdot)$ is a rounding function and $\text{clip}(\cdot, -B, B)$ clamps the input into the range $[-B, B]$. The quantizer $Q_{\delta,B}(x)$ converts $x$ to quantities within the range $[-\delta B, \delta B]$, up to the numerical precision $\delta$. The quantized value can be stored as a $\log_2(2B)$-bit integer. The quantizer can be generalized to vector inputs and outputs by applying element-wise to each entry.

*Arithmetic underflow* occurs when we try to sum up two quantities of different orders of magnitude. Consider the summation

$$\theta_t = Q_{\delta,B}\left(\theta_{t-1} + v_t\right), \quad (4)$$

where $\theta_t$ is the parameter at the $t$-th iteration, and $v_t = -\alpha m_t$ is the update vector. Since $\theta_{t-1}$ itself is already in low-precision, we have $Q_{\delta,B}(\theta_{t-1}) = \theta_{t-1}$. Now, if the update is smaller than parameters' numerical precision, *i.e.* $|v_{t,d}| < \delta/2$ for some dimension $d$, the small update will be "eaten" by the large parameter, treated effectively as zero:

$$\theta_{t,d} = Q_{\delta,B}\left(\theta_{t-1,d} + v_{t,d}\right) = \theta_{t-1,d}. \quad (5)$$

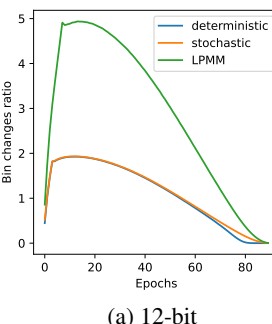 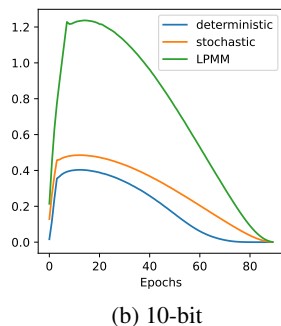 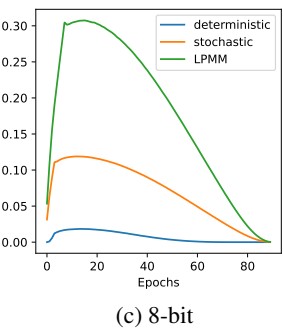

(a) 12-bit            (b) 10-bit            (c) 8-bit

Figure 2: The magnitude of parameter update for a convolution layer in ResNet-18. We vary the parameter precision from 12-bit to 8-bit. Gradients and momentum are always in 8-bit. The "bin change ratio" is defined as $\tilde{v}_t/\delta$ averaged over all dimensions. Larger is better.

The update is likely to be small, particularly at the late stage of training where the gradient (and momentum) is close to zero, and the learning rate is also decreased around zero. If underflowing occurs, the actual update direction $\tilde{v}_t := \theta_t - \theta_{t-1}$ is far from the expected. the Extremely, there could be $\tilde{v}_t = 0$, where the optimization completely stalls.

As shown in Fig. 2, when the underflow problem is serious (fewer bits), the deterministic quantizer has the smallest actual update $\tilde{v}_t/\delta$. Actually, the deterministic quantizer hardly updates the parameters. As shown in Fig. 3a, about 60% entries in the update vector is less than $\delta/2$. These entries are ignored by the deterministic quantizer.

### 3.3 SOLUTIONS TO THE UNDERFLOW PROBLEM

We propose two methods for solving the underflow problem: unbiased stochastic quantization and gradient accumulation.

**Unbiased Stochastic Quantization** Stochastic quantization adopts an stochastic quantizer $Q(\cdot)$, satisfying $\mathbb{E}\left[Q(x)\right] = x$. A simplest stochastic quantizer is the following:

$$Q_{\delta,B}(x) = \delta \cdot \text{SR}\left(\text{clip}\left(x/\delta, -B, B\right)\right), \tag{6}$$

where $\text{SR}(\cdot)$ is the stochastic rounding (Gupta et al., 2015; Courbariaux et al., 2015) operation:

$$\text{SR}(x) = \lceil x \rceil \quad \text{with probability } x - \lfloor x \rfloor \text{ otherwise } \lfloor x \rfloor.$$

The quantizer is unbiased as long as $\delta B \geq |x|$. Stochastic quantization mitigates underflow by accumulating small updates in a probabilistic manner. When the update is small in magnitude, each dimension only has a small probability to be updated. Yet after a large number of updates, the final parameters still have the same expectation with the full-precision ones. Besides this informal argument, unbiased stochastic quantization is also the key to establish the convergence theory, as we shall see soon in Sec. 4.2.

**Microbatching** Underflowing arises from the difference in magnitude of the parameter and the update. Therefore, it can be mitigated by enlarging the update. *Microbatching* (Huang et al., 2019) works by summing up the gradient of multiple batches before applying the update to the momentum and parameters. More concretely, we can accumulate the gradient of $N$ mini-batches as

$$g_{t,0} \leftarrow 0, \quad g_{t,i} \leftarrow Q(g_{t,i-1} + \nabla f(\theta_{t-1}, \zeta^{t,i})),$$

and use $g_{t,N}$ in place of $g_t$ for updating the momentum in Eq. (2). Now, the accumulated gradient $g_{t,N}$ is $N$ times larger than the original $g_t$. Subsequently, the update $-\alpha m_t$ becomes $N$ times larger, yet the magnitude of the parameter $\theta_t$ is unchanged, so underflowing is significantly alleviated.

As shown in Fig. 3(a), about 20% entries of update vector $v_t$ are even smaller than $\delta/8$. In this case, vanilla stochastic quantizer can still suffer from high variance due to the repeated small updates.

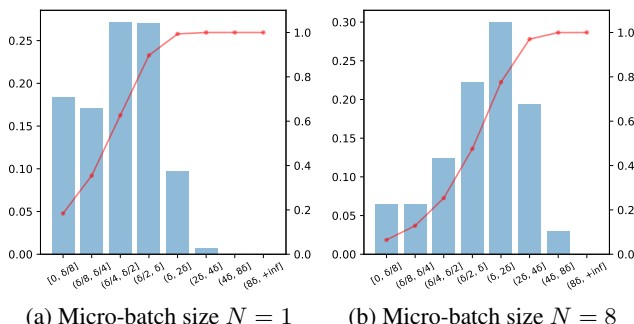

(a) Micro-batch size $N = 1$      (b) Micro-batch size $N = 8$

Figure 3: The distribution of update vector magnitude across dimensions. The result is counted at the epoch 10 for a single layer in ResNet-18. The blue bar (left Y-axis) represents the density and the red line (right Y-axis) indicates the cumulative distribution.

Microbatching significantly enlarge the size the update, and reduce the number of small updates (which underflow), as shown in Fig. 3(b).

Micro-batching can be viewed equivalently as large-minibatch SGD (Goyal et al., 2017), which simply multiply the batch size and the learning rate by $N$. Large-minibatch SGD is known to converge similarly to the original, small-minibatch SGD. However, enlarging the batch size is not memory efficient since the data memory will increase.

Putting it all together, we propose LPMM-SGDM, an optimizer with low-precision gradient accumulators, momentum, and parameters. The entire training procedure is given by Algorithm 1. Fig. 2 and 3 show that our algorithm effectively mitigate underflow with significantly larger updates.

---

**Algorithm 1** LPMM-SGDM

---

**Require:** initial parameter $\theta_0 \in \mathbb{R}^p$, learning rate $\alpha$, initial momentum $m_0 = 0 \in \mathbb{R}^p$, quantization function $Q$, total number of iterations $T$, momentum factor $\beta$, gradient accumulation steps $N$
1: **for** $t = 1, 2, \ldots, T$ **do**
2:     $g_{t,0} \leftarrow 0$
3:     **for** $i = 1, 2, \ldots, N$ **do**
4:         Sample a minibatch $\zeta^{t,i}$ uniformly from the training data;
5:         $g_{t,i} \leftarrow Q(g_{t,i-1} + \nabla f(\theta_{t-1}, \zeta^{t,i}))$
6:     **end for**
7:     $m_t \leftarrow Q(\beta m_{t-1} + g_{t,N}))$
8:     $\theta_t \leftarrow Q(\theta_{t-1} - \alpha m_t)$
9: **end for**
10: **return** $\theta_T$

---

## 4   CONVERGENCE THEORY

In this section, we analyze the convergence of the proposed LPMM-SGDM algorithm and discuss the impact of the proposed techniques.

### 4.1   ASSUMPTIONS

First, we make some assumptions. The first three are rather standard in stochastic optimization literature, while last two depict properties of stochastic quantizers.

1. *(Convexity) The objective function is convex and has an unique global minimum $f(\theta^*)$.*
2. *(Smoothness) The objective $f(\theta)$ is continuous differentiable and L-smooth;*

3. *(Bounded variance)* $\mathbb{E}\left[\left\|\nabla\tilde{f}(\theta) - \nabla f(\theta)\right\|^2\right] < \sigma^2, \forall \theta \in \mathbb{R}^d$.

4. *(Unbiased quantizer)* $\forall x \in \mathbb{R}^d, \mathbb{E}[Q_p(x)] = \mathbb{E}[Q_g(x)] = \mathbb{E}[Q_m(x)] = x$.

5. *(Bounded quantization variance)* $\forall x \in \mathbb{R}^d, \mathbb{E}\left[\|Q_p(x) - x\|^2\right] \le \sigma_p^2, \mathbb{E}\left[\|Q_g(x) - x\|^2\right] \le \sigma_g^2,$
   $\mathbb{E}\left[\|Q_m(x) - x\|^2\right] \le \sigma_m^2$.

Here, Assumption 3 depicts the variance of sampling mini-batches, and Assumption 5 bounds quantization error of model-related statistics through the variance. Assumptions 4-5 can be realized by the linear quantizer in Eq. (6), where we have $\mathbb{E}\left[\|Q_{\delta,B}(x) - x\|^2\right] \le \frac{\delta^2 d}{4}, \forall x \in \mathbb{R}^d$. Yet, our analysis is applicable to other non-linear quantizers such as those discussed in Sec. 5.1. We assume that the objective function is convex, which is somewhat restrictive. However, this simple convex case is sufficient to reflect the underflowing challenge of training with low-precision model memory, and we leave the non-convex case as a future study.

## 4.2 CONVERGENCE OF LPMM-SGDM

We first analyze the bias and variance of the quantized gradient accumulator.

**Lemma 1.** *In Algorithm 1, with Assumptions 3-5, the first and second momentum of $g_{t,i}$ satisfy*

$$\mathbb{E}[g_{t,i}|\theta_{t-1}] = i\nabla f(\theta_{t-1}) \tag{7}$$

$$\mathbb{E}[\|g_{t,i}\|^2|\theta_{t-1}] \le i\sigma_g^2 + i(i+2)\|\nabla f(\theta_{t-1})\|^2 + i\sigma^2. \tag{8}$$

Next, we construct an auxiliary sequence $\{z_t\}$, following previous analysis of heavy ball methods (Ghadimi et al., 2015) and SGDM (Liu et al., 2020):

$$z_t = \theta_0, \text{ when } t = 0, \text{ otherwise } \frac{1}{1-\beta}\theta_t - \frac{\beta}{1-\beta}\theta_{t-1}. \tag{9}$$

The sequence enjoys the following nice properties:

**Lemma 2.** *If Assumptions 3-5 hold, then sequence $\{z_t\}$ satisfies*

$$z_{t+1} - z_t = \frac{1}{1-\beta}(\theta_{t+1} - \theta_t) - \frac{\beta}{1-\beta}(\theta_t - \theta_{t-1}) \tag{10}$$

$$\mathbb{E}[z_{t+1} - z_t] = \frac{-\alpha N}{1-\beta}\nabla f(\theta_t) \tag{11}$$

$$\mathbb{E}[\|z_{t+1} - z_t\|^2] \le \left(\frac{\alpha}{1-\beta}\right)^2 \left(2\mathbb{E}[\|g_{t+1,N}\|^2] + \sigma_m^2\right) + \frac{1+2\beta^2}{(1-\beta)^2}\sigma_p^2. \tag{12}$$

The quantity $\mathbb{E}[\|z_{t+1} - z_t\|^2]$ is composed of four parts: the batch gradient norm, the variance of stochastic gradient, momentum quantization and parameter quantization error. The latter three noise terms sum up independently. Our analysis applies to cases when only some of the model-related statistics are quantized by setting corresponding noise terms to zero. With this lemma, we derive the following theorem characterizing the convergence of LPMM-SGDM.

**Theorem 1.** *Consider the Algorithm 1 with Assumptions 1-5. Let $\alpha \in (0, \frac{1-\beta}{(N+2)L}]$, then for all $T > 0$ we have*

$$\mathbb{E}[f(\bar{\theta}_T) - f_*] \le \frac{1}{2T}\left(\frac{L\beta}{1-\beta} + \frac{1-\beta}{\alpha N}\right)\|\theta_0 - \theta_*\|^2 \tag{13}$$

$$+ \frac{\alpha\sigma^2}{(1-\beta)} + \frac{\alpha\sigma_g^2}{(1-\beta)} + \frac{\alpha\sigma_m^2}{2N(1-\beta)} + \frac{(1+2\beta^2)\sigma_p^2}{2\alpha N(1-\beta)}. \tag{14}$$

*where $\bar{\theta}_T = \frac{1}{T}\sum_{i=0}^{T-1}\theta_i$.*

Theorem 1 reveals the effectiveness of the proposed techniques in Sec. 3.3. First, LPMM can converge to the global optimum when $T \to \infty$, $\alpha \to 0$, and $\alpha N \to \infty$ for any quantizer with non-zero

$\sigma_p$, $\sigma_g$, and $\sigma_m$. That is, the ultimate convergence is guaranteed regardless of the numerical precision. In contrast, optimizers with deterministic quantizers could stall at some intermediate state, as discussed in Sec. 3.3. Therefore, stochastic quantization is essential for the convergence guarantee.

Second, we can study the impact of gradient accumulation. With sufficiently large $T$, the first term in Eq. (14) vanishes, and we therefore only consider the noise terms. We compare a plain algorithm ($N = 1$) and a version with gradient accumulation ($N > 1$), with equal learning rate. Gradient accumulation reduces the parameter quantization error (the $\sigma_p$ term) by $N$ times. Meanwhile, the gradient accumulator stores the gradient sum over $N$ mini-batches, so $\sigma_g$ becomes larger. Therefore, gradient accumulation trades gradient quantization error for parameter quantization error. The error bound is tighter with a large $N$ when parameter quantization error is more significant, which holds true according to our empirical observations in Fig. 2 and 3.

## 5 Implementation Details

In this section, we discuss our quantizer design and the implementation of the whole training framework.

### 5.1 Group and Nonlinear Quantization

For the quantizer, we use the nonlinear quantization on momentum and linear quantization on parameters and gradients. All quantizers use the standard absmax symmetric quantization. The linear quantization is defined as Eq. 6. The nonlinear quantization method (Dettmers, 2015) proposes a novel floating-point format in which the bitwidth for the exponent is dynamic. Furthermore, given a tensor to be quantized, we first use the absmax to normalize the tensor into $[-1, 1]$. Then, the quantizer finds the bucket for each element with a binary search on a precomputed quantization map. Finally, the quantizer projects the value to one of the adjacent quantization points by stochastic rounding. Since stochastic rounding is unbiased, this nonlinear quantization method is also unbiased, which indicates that all the analysis above holds except the quantization variance is intrinsically different.

Group quantization is also adopted to avoid the negative impact of outliers. We chunk every tensor into groups of size 2048, and every group has its own quantization precision $\delta$. However, group quantization introduces more data to be stored and as group size decreases, the quantization is more accurate while the extra data cost will be higher. We choose the group size of 2048 to balance the numerical precision and storage cost.

### 5.2 Dynamic Quantization

We experimentally observed that the magnitude of parameters shows a trend of first increasing and then decreasing. If a smaller quantization granularity $\delta$ is adopted, then the *overflow* problems occur. Especially at the initial stage of optimization, the quantizer rudely truncated the parameters to be increased due to limited range, which leads to a biased quantizer and makes the solution suboptimal. However, if we quantize with an overly large $\delta$, it results in poor numerical precision, and the underflow problem worsens. Therefore, we take a dynamic quantization policy to balance the overflow and underflow. Specifically, it calculates the quantization precision $\delta$ with the quantizers' inputs at every iteration, which gives a minimal lossless $\delta$.

### 5.3 System Implementation

Here we discuss how our algorithm is implemented. As illustrated in Fig. 1, the forward, backward, and optimizer steps are all depicted straightforwardly. Before an iteration begins, we store all model memory in low precision. When the new input comes, the parameters needed for forward computation are dequantized into a temporary full-precision "copy". Note that the parameters for each layer are not required simutaenously, so we only dequantize parameters right before its usage. As long as the computation related to the full-precision "copy" has finished, we delete these tensors. Therefore, the overhead of the temporary full-precision parameter is no larger than the largest single parameter tensor. In the backpropagation and optimizer step, a similar computation pattern happens. Note that the memory of gradient accumulators cannot be saved by not storing the gradient accumulator even

Table 1: Performance of ResNet-18 and ResNet50 on ImageNet.

| Model | Method | Bits (param./grad./moment.) | Accuracy(%) |
|-----------|---------------|:-----------------------------:|:-------------:|
| ResNet-18 | Full-Precsion | 32/32/32 | 71.1 |
| ResNet-18 | LPMM | 12/8/8 | 71.0 |
| ResNet-50 | Full-Precsion | 32/32/32 | 77.2 |
| ResNet-50 | LPMM | 12/8/8 | **77.3** |

if we do not apply microbatching (*i.e.* $N = 1$). On the one hand, the gradient may not be calculated in one operation (or one layer). On the other hand, gradient computation can be composed of multiple accumulations, such as in a convolution layer.

Since the computation in neural network training is always layerwise, then at most, only the full-precision parameters, gradients, and momentum associated with one single layer are kept in memory at the same time, while other model memory is inactive and in low-precision.

Based on this philosophy, we implemented a training algorithm on PyTorch. We quantize the full-precision to low-precision representation by manually compressing them into a byte stream using the CUDA kernel. Our system includes a lot of operators and layers to perform desired forward and backward calculations. Given a defined model, we quantize the parameter tensor individually, replace the original full-precision parameter and then move to GPU. Then it can convert the layers of the model automatically by our customized layer. Besides, to get the gradient and perform computation more conveniently, we manually design a gradient accumulator instead of using the original "grad" attribute.

## 6 EXPERIMENTS

**Experimental Setup**   We compare the performance of LPMM to the full-precision counterparts on the image classification task. The full-precision baseline uses the standard SGDM optimizer (Qian, 1999; Sutskever et al., 2013), and we use our proposed LPMM-SGDM algorithm. We experiment with ResNet-18 and ResNet-50 (He et al., 2016) on ImageNet (Deng et al., 2009). We do not change any existing hyperparameters or precision of input and activations for each experimental run. The only change is to replace full-precision parameters, gradients, and momentum with the low-precision ones. We conduct ResNet-18 experiments on RTX 3090Ti and ResNet-50 experiments on A100 GPUs.

The accuracy result is shown in Tab. 1. LPMM-SGDM can quantize gradients and the momentum to 8-bit and model parameters to 12-bit, with no accuracy loss compared with the full precision SGDM.

**Ablation Study**   To study the effectiveness of our proposed techniques, we conduct an ablation analysis on ResNet-18 on LPMM under 12-bit, 10-bit, and 8-bit parameter quantization. We consistently use 8-bit gradients and 8-bit momentum. The result is shown in Tab. 2. The baseline uses a deterministic dynamic quantizer (rounding to the nearest), and no microbatching is applied. We then apply stochastic quantization and microbatching sequentially. The performance with microbatching is the optimal result across different micro-batch size, which is $N = 8$ for 12-bit and $N = 16$ for 10-bit and 8-bit. The results show that stochastic rounding and microbatching are critical for the performance of LPMM. The 8-bit runs collapsed without the unbiasedness guarantee. While as the bit goes up, the reduced quantization error compensates for this bias. Microbatching is helpful at all quantization level and make the 12-bit parameter LPMM match with full-precision.

**Role of Stochastic Quantization**   We compare the effect of stochastic rounding on LPMM-SGDM across 12-bit, 10-bit, and 8-bit parameter quantization. In these experiments, "deterministic" means all quantization operations use the "rounding to the nearest" quantizer, while "stochastic" means all quantization operations are stochastic and no microbatching is applied. The result is shown in Fig. 4. For the 12-bit setting, we can see that the training trajectories overlap in the early stage of optimization since the update is large enough compared to the numerical precision $\delta$, and underflowing is not significant. Nevertheless, the deterministic method degrades from the middle stage due to the smaller update vector, which leads to underflowing. The degradation occurs earlier for 10-bit

Table 2: Ablation study of LPMM-SGDM on ResNet-18 on ImageNet.

| Method | Bits (param./grad./moment.) | Baseline | +Stochastic | +Microbatching |
|---|---|---|---|---|
| Full-Precision | 32/32/32 | 71.1 | - | - |
| LPMM | 12/8/8 | 66.0 | 70.7 | **71.0** |
| LPMM | 10/8/8 | 53.4 | 69.7 | **70.6** |
| LPMM | 8/8/8 | 15.8 | 65.6 | **69.2** |

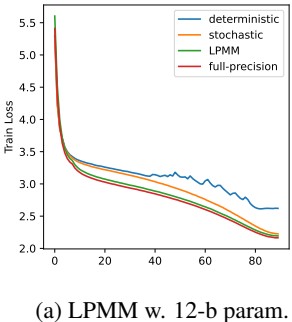 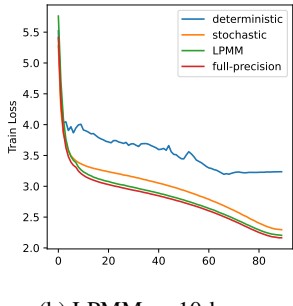 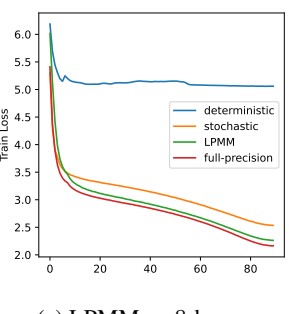

(a) LPMM w. 12-b param.  (b) LPMM w. 10-b param.  (c) LPMM w. 8-b param.

Figure 4: Convergence curves of compared algorithms.

and 8-bit settings since the quantization precision $\delta$ is larger, and underflow can happen more easily. This indicates that unbiased stochastic quantization is the crucial factor in the underflow problem.

**Role of Micro-batching** Here, we study the impact of the micro-batch size $N$. In these experiments, we use a micro-batch size of 256 and an initial learning rate 0.256. Other hyperparameters such as warmup and weight decay are set according to the actual batch size (i.e., $256N$). The result is shown in Tab. 3. When the parameter quantization error is large, particularly in 8-bit setting, microbatching mitigates the parameter quantization error effectively while the increased gradient error is still tolerable. We can see that microbatching exactly plays a role on reducing parameter quantization error since the improvement on 8-bit setting is largest. The improvement on 12-bit is marginal since the parameter quantization error is already small.

Table 3: Effect of the micro-batch size.

| Micro-Batch Size($N$) | 1 | 2 | 4 | 8 | 16 |
|---|---|---|---|---|---|
| Full-Precision | 71.1 | - | - | - | - |
| LPMM (12-b param. / 8-b grad. / 8-b moment.) | 70.7 | 70.0 | 70.5 | **71.0** | 70.9 |
| LPMM (10-b param. / 8-b grad. / 8-b moment.) | 69.7 | 69.0 | 70.1 | 70.3 | **70.6** |
| LPMM (8-b param. / 8-b grad. / 8-b moment.) | 65.6 | 65.9 | 68.1 | 68.8 | **69.4** |

## 7 CONCLUSIONS

We present LPMM, an optimization framework for training neural networks with the entire model memory kept in low-precision. We apply LPMM to the SGDM optimizer and give convergence guarantees. We discover the critical underflow problem in low-precision training and propose two methods: unbiased stochastic quantization and microbatching, as the solution for this problem. Moreover, we conduct a comprehensive study on the effect of stochastic and microbatching theoretically and empirically. On the image classification task, LPMM achieves an accuracy with negligible loss using 12-bit parameters, 8-bit gradients, and 8-bit momentum, which reduce over 70% model memory compared with the full-precision training.

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

# A   PROOF OF LEMMA AND THEOREM

## A.1   PROOF OF LEMMA 1

*Proof.* When $i = 1$, by Assumption 4 and the due to $\nabla \tilde{f}(\theta)$ is unbiased, we can get

$$
\begin{aligned}
\mathbb{E}\left[g_{t,1}|\theta_{t-1}\right] &= \mathbb{E}\left[Q(\nabla \tilde{f}(\theta_{t-1}))\right] \\
&= \mathbb{E}\left[\nabla \tilde{f}(\theta_{t-1})\right] \\
&= \nabla f(\theta_{t-1})
\end{aligned}
$$

And when $i > 1$ we have

$$
\begin{aligned}
\mathbb{E}\left[g_{t,i}|\theta_{t-1}\right] &= \mathbb{E}\left[Q(g_{t,i-1} + \nabla \tilde{f}(\theta_{t-1}))|\theta_{t-1}\right] \\
&= \mathbb{E}\left[(g_{t,i-1} + \nabla \tilde{f}(\theta_{t-1}))|\theta_{t-1}\right] \\
&= \mathbb{E}\left[g_{t,i-1}|\theta_{t-1}\right] + \mathbb{E}\left[\nabla \tilde{f}(\theta_{t-1})\right] \\
&= \mathbb{E}\left[g_{t,i-1}|\theta_{t-1}\right] + \nabla f(\theta_{t-1})
\end{aligned}
$$

By reduction it gives the first part of Lemma 1

$$
\mathbb{E}\left[g_{t,i}|\theta_{t-1}\right] = i\nabla f(\theta_{t-1})
$$

With Assumption 3, it holds true that

$$
\begin{aligned}
\mathbb{E}\left[\left\|\nabla \tilde{f}(\theta)\right\|^2\right] &= \mathbb{E}\left[\left\|(\nabla \tilde{f}(\theta) - \nabla f(\theta)) + \nabla f(\theta)\right\|^2\right] \\
&= \mathbb{E}\left[\left\|\nabla \tilde{f}(\theta) - \nabla f(\theta)\right\|^2\right] + \mathbb{E}\left[\|\nabla f(\theta)\|^2\right] - 2\mathbb{E}\left[< \nabla \tilde{f}(\theta) - \nabla f(\theta), \nabla f(\theta) >\right] \\
&= \mathbb{E}\left[\left\|\nabla \tilde{f}(\theta) - \nabla f(\theta)\right\|^2\right] + \mathbb{E}\left[\|\nabla f(\theta)\|^2\right] \\
&< \sigma^2 + \|\nabla f(\theta)\|^2
\end{aligned}
$$

For the second part of Lemma 1, when $i = 1$,

$$
\begin{aligned}
\mathbb{E}\left[\|g_{t,1}\|^2 |\theta_{t-1}\right] &= \mathbb{E}\left[\left\|g_{t,1} - \nabla \tilde{f}(\theta_{t-1}) + \nabla \tilde{f}(\theta_{t-1})\right\|^2 |\theta_{t-1}\right] \\
&= \mathbb{E}\left[\left\|g_{t,1} - \nabla \tilde{f}(\theta_{t-1})\right\|^2\right] + \mathbb{E}\left[\left\|\nabla \tilde{f}(\theta_{t-1})\right\|^2\right] - 2\mathbb{E}\left[< g_{t,1} - \nabla \tilde{f}(\theta_{t-1}), \nabla \tilde{f}(\theta_{t-1}) >\right] \\
&= \mathbb{E}\left[\left\|g_{t,1} - \nabla \tilde{f}(\theta_{t-1})\right\|^2\right] + \mathbb{E}\left[\left\|\nabla \tilde{f}(\theta_{t-1})\right\|^2\right] \\
&\leq \sigma_g^2 + \|\nabla f(\theta_{t-1})\|^2 + \sigma^2
\end{aligned}
$$

And when $i > 1$, if this equation holds true for 1 to $i - 1$, then for $i$ we have

$$
\begin{aligned}
\mathbb{E}\left[\|g_{t,i}\|^2 \,|\theta_{t-1}\right] &= \mathbb{E}\left[\left\|g_{t,i} - g_{t,i-1} - \nabla\tilde{f}(\theta_{t-1})\right\|^2\right] + \mathbb{E}\left[\left\|g_{t,i-1} + \nabla\tilde{f}(\theta_{t-1})\right\|^2\right] \\
&\quad - 2\mathbb{E}\left[< g_{t,i} - g_{t,i-1} - \nabla\tilde{f}(\theta_{t-1}), g_{t,i-1} + \nabla\tilde{f}(\theta_{t-1}) >\right] \\
&= \mathbb{E}\left[\left\|g_{t,i} - g_{t,i-1} - \nabla\tilde{f}(\theta_{t-1})\right\|^2\right] + \mathbb{E}\left[\left\|g_{t,i-1} + \nabla\tilde{f}(\theta_{t-1})\right\|^2\right] \\
&\le \sigma_g^2 + \mathbb{E}\left[\|g_{t,i-1}\|^2\right] + \mathbb{E}\left[\left\|\nabla\tilde{f}(\theta_{t-1})\right\|^2\right] + 2\mathbb{E}\left[< g_{t,i-1}, \nabla\tilde{f}(\theta_{t-1})\right] \\
&= \sigma_g^2 + \mathbb{E}\left[\|g_{t,i-1}\|^2\right] + \mathbb{E}\left[\left\|\nabla\tilde{f}(\theta_{t-1})\right\|^2\right] + 2(i-1)\|\nabla f(\theta_{t-1})\|^2 \\
&\le \sigma_g^2 + (i-1)\sigma_g^2 + (i-1)(i+1)\|\nabla f(\theta_{t-1})\|^2 + (i-1)\sigma^2 + \|\nabla f(\theta_{t-1})\|^2 + \sigma^2 \\
&= i\sigma_g^2 + (i^2 + 2i - 2)\|\nabla f(\theta_{t-1})\|^2 + i\sigma^2 \\
&\le i\sigma_g^2 + i(i+2)\|\nabla f(\theta_{t-1})\|^2 + i\sigma^2
\end{aligned}
$$

So by reduction we complete the second part. $\qquad\square$

## A.2 PROOF OF LEMMA 2

*Proof.* By definition of $z_t$, we have the first equation immediately. With it we get

$$
\mathbb{E}[z_{t+1} - z_t] = \frac{1}{1-\beta}\mathbb{E}[\theta_{t+1} - \theta_t] - \frac{\beta}{1-\beta}\mathbb{E}[\theta_t - \theta_{t-1}]
$$

By Algorithm 1 we have

$$
\begin{aligned}
\mathbb{E}[\theta_{t+1} - \theta t] &= \mathbb{E}[\theta_{t+1} - (\theta t - \alpha m_{t+1})] - \mathbb{E}[\alpha m_{t+1}] \\
&= -\alpha\mathbb{E}[m_{t+1}] \\
&= -\alpha\mathbb{E}[\beta m_t - g_{t+1,N}] \\
&= -\alpha\beta\mathbb{E}[m_t] - \alpha N\nabla f(\theta_t)
\end{aligned}
$$

and

$$
\begin{aligned}
\mathbb{E}[\theta_t - \theta_{t-1}] &= \mathbb{E}[\theta_t - (\theta_{t-1} - \alpha m_t)] - \mathbb{E}[\alpha m_t] \\
&= -\alpha\mathbb{E}[m_t]
\end{aligned}
$$

Substitute them we derive the second equation

$$
\begin{aligned}
\mathbb{E}[z_{t+1} - z_t] &= -\frac{\alpha\beta}{1-\beta}\mathbb{E}[m_t] - \frac{\alpha N}{1-\beta}\nabla f(\theta_t) + \frac{\alpha\beta}{1-\beta}\mathbb{E}[m_t] \\
&= \frac{-\alpha N}{1-\beta}\nabla f(\theta_t)
\end{aligned}
$$

As for the last part, since

$$
\begin{aligned}
z_{t+1} - z_t &= \frac{1}{1-\beta}(\theta_{t+1} - \theta_t) - \frac{\beta}{1-\beta}(\theta_t - \theta_{t-1}) \\
&= \frac{1}{1-\beta}(\theta_{t+1} - (\theta_t - \alpha m_{t+1})) - \frac{\beta}{1-\beta}(\theta_t - (\theta_{t-1} - \alpha m_t)) - \frac{\alpha}{1-\beta}(m_{t+1} - \beta m_t)
\end{aligned}
$$

Take expectation and we have

$$\mathbb{E}\left[\|z_{t+1} - z_t\|^2\right] = \mathbb{E}\left[\left\|\frac{1}{1-\beta}(\theta_{t+1} - (\theta_t - \alpha m_{t+1})) - \frac{\beta}{1-\beta}(\theta_t - (\theta_{t-1} - \alpha m_t)) - \frac{\alpha}{1-\beta}(m_{t+1} - \beta m_t)\right\|^2\right]$$

$$= \frac{1}{(1-\beta)^2}\mathbb{E}\left[\|\theta_{t+1} - (\theta_t - \alpha m_{t+1})\|^2\right] + \frac{\beta^2}{(1-\beta)^2}\mathbb{E}\left[\|\theta_t - (\theta_{t-1} - \alpha m_t)\|^2\right]$$

$$+ \left(\frac{\alpha}{1-\beta}\right)^2 \mathbb{E}\left[\|m_{t+1} - \beta m_t\|^2\right]$$

$$- \frac{2\beta}{(1-\beta)^2}\mathbb{E}\left[< \theta_{t+1} - (\theta_t - \alpha m_{t+1}), \theta_t - (\theta_{t-1} - \alpha m_t) >\right]$$

$$- \frac{2\alpha}{(1-\beta)^2}\mathbb{E}\left[< \theta_{t+1} - (\theta_t - \alpha m_{t+1}), m_{t+1} - \beta m_t >\right]$$

$$+ \frac{2\alpha\beta}{(1-\beta)^2}\mathbb{E}\left[< \theta_t - (\theta_{t-1} - \alpha m_t), m_{t+1} - \beta m_t >\right]$$

$$\leq \frac{1+\beta^2}{(1-\beta)^2}\sigma_p^2 + \left(\frac{\alpha}{1-\beta}\right)^2 \left(\mathbb{E}\left[\|m_{t+1} - (\beta m_t + g_{t+1,N})\|^2\right]\right.$$

$$+ \mathbb{E}\left[\|g_{t+1,N}\|^2\right] - 2\mathbb{E}\left[< m_{t+1} - (\beta m_t + g_{t+1,N}), g_{t+1,N} >\right])$$

$$\leq \left(\frac{\alpha}{1-\beta}\right)^2 \left(\mathbb{E}\left[\|g_{t+1,N}\|^2\right] + \sigma_m^2\right) + \frac{1+\beta^2}{(1-\beta)^2}\sigma_p^2$$

which is the last part of Lemma 2. □

### A.3 PROOF OF THEOREM 1

*Proof.* From Lemma 2, we have

$$\mathbb{E}\left[\|z_{t+1} - z_t\|^2\right] \leq \left(\frac{\alpha}{1-\beta}\right)^2 \left(2\mathbb{E}\left[\|g_{t+1,N}\|^2\right] + \sigma_m^2\right) + \frac{1+2\beta^2}{(1-\beta)^2}\sigma_p^2$$

And by Lemma 1, we know

$$\mathbb{E}\left[\|g_{t+1,N}\|^2\right] \leq N\sigma_g^2 + N(N+2)\|\nabla f(\theta_t)\|^2 + N\sigma^2$$

Substituting it gives

$$\mathbb{E}\left[\|z_{t+1} - z_t\|^2\right] \leq \left(\frac{\alpha}{1-\beta}\right)^2 \left(2N(N+2)\|\nabla f(\theta_t)\|^2 + 2N\sigma^2 + 2N\sigma_g^2 + \sigma_m^2\right) + \frac{1+2\beta^2}{(1-\beta)^2}\sigma_p^2 \tag{15}$$

Suppose $\theta_*$ is the optimal parameter and $f_* = f(\theta_*)$ is the minimal objective value. First, we have

$$\|z_{t+1} - \theta_*\|^2 = \|z_t - \theta_*\|^2 + 2\langle z_t - \theta_*, z_{t+1} - z_t\rangle + \|z_{t+1} - z_t\|^2$$

Take expectation over the randomness in the (t+1)-th step, we have

$$\mathbb{E}[\|z_{t+1} - \theta_*\|^2] = \|z_t - \theta_*\|^2 - \frac{2\alpha N}{1-\beta}\langle z_t - \theta_*, \nabla f(\theta_t)\rangle + \mathbb{E}[\|z_{t+1} - z_t\|^2]$$

$$= \|z_t - \theta_*\|^2 - \frac{2\alpha N}{1-\beta}\langle \theta_t - \theta_*, \nabla f(\theta_t)\rangle$$

$$- \frac{2\alpha\beta N}{(1-\beta)^2}\langle \theta_t - \theta_{t-1}, \nabla f(\theta_t)\rangle + \mathbb{E}[\|z_{t+1} - z_t\|^2]$$

Since $f$ is continuously differentiable and L-smooth, we have the following inequalities. (Nesterov, 2013)

$$\langle \theta_t - \theta_*, \nabla f(\theta_t)\rangle \geq \frac{1}{L}\|\nabla f(\theta_t)\|^2 \tag{16}$$

$$\langle \theta_t - \theta_*, \nabla f(\theta_t)\rangle \geq f(\theta_t) - f_* + \frac{1}{2L}\|\nabla f(\theta_t)\|^2 \tag{17}$$

$$\langle \theta_t - \theta_{t-1}, \nabla f(\theta_t)\rangle \geq f(\theta_t) - f(\theta_{t-1}) \tag{18}$$

Substitute them and get

$$\mathbb{E}[\|z_{t+1} - \theta_*\|^2] \le \|z_t - \theta_*\|^2 - \frac{2\alpha N(1-\rho)}{L(1-\beta)}\|\nabla f(\theta_t)\|^2 - \frac{2\alpha N\rho}{1-\beta}(f(\theta_t) - f_*)$$

$$\frac{\alpha N\rho}{L(1-\beta)}\|\nabla f(\theta_t)\|^2 - \frac{2\alpha N\beta}{(1-\beta)^2}(f(\theta_t) - f(\theta_{t-1})) + \mathbb{E}[\|z_{t+1} - z_t\|^2]$$

where $\rho \in (0, 1]$ is a parameter we use to balance the first two inequalities. For the last term, denote $M = \left(\frac{\alpha}{1-\beta}\right)^2 (2N\sigma^2 + 2N\sigma_g^2 + N\sigma_m^2) + \frac{1+2\beta^2}{(1-\beta)^2}\sigma_p^2$. Substitute 15 into the last inequality and collect the terms, we get

$$\left(\frac{2\alpha N\rho}{1-\beta} + \frac{2\alpha N\beta}{(1-\beta)^2}\right)(f(\theta_t) - f_*) + \mathbb{E}[\|z_{t+1} - \theta_*\|^2]$$

$$\le \frac{2\alpha N\beta}{(1-\beta)^2}(f(\theta_{t-1}) - f_*) + \|z_t - \theta_*\|^2 + \left(\frac{2\alpha^2 N(N+2)}{(1-\beta)^2} - \frac{\alpha N(2-\rho)}{L(1-\beta)}\right)\|\nabla f(\theta_t)\|^2 + M$$

When $\alpha$ satisfies the condition $\frac{2\alpha^2 N(N+2)}{(1-\beta)^2} - \frac{\alpha N(2-\rho)}{L(1-\beta)} \le 0$, i.e. $0 \le \alpha \le \frac{(1-\beta)(2-\rho)}{2(N+2)L}$, the term about $\|\nabla f(\theta_t)\|^2$ is non-positive, thus we have

$$\left(\frac{2\alpha N\rho}{1-\beta} + \frac{2\alpha N\beta}{(1-\beta)^2}\right)(f(\theta_t) - f_*) + \mathbb{E}[\|z_{t+1} - \theta_*\|^2]$$

$$\le \frac{2\alpha N\beta}{(1-\beta)^2}(f(\theta_{t-1}) - f_*) + \|z_t - \theta_*\|^2 + M$$

Summing this inequality from 0 to $T-1$ and taking full expectation gives

$$\frac{2\alpha N\rho}{1-\beta}\sum_{i=0}^{T-1}\mathbb{E}[f(\theta_i) - f_*] + \sum_{i=0}^{T-1}\left(\frac{2\alpha N\beta}{(1-\beta)^2}\mathbb{E}[f(\theta_i) - f_*] + \mathbb{E}[\|z_{i+1} - \theta_*\|^2]\right)$$

$$\le \sum_{i=0}^{T-1}\left(\frac{2\alpha N\beta}{(1-\beta)^2}\mathbb{E}[f(\theta_{i-1}) - f_*] + \mathbb{E}[\|z_i - \theta_*\|^2]\right) + T \cdot M$$

which implies that

$$\frac{2\alpha N\rho}{1-\beta}\sum_{i=0}^{T-1}\mathbb{E}[f(\theta_i) - f_*] \le \frac{2\alpha N\beta}{(1-\beta)^2}(f(\theta_0) - f_*) + \|\theta_0 - \theta_*\|^2 + T \cdot M$$

Since $f$ is **convex**, we have $Tf(\bar\theta_T) \le \frac{1}{T}\sum_{i=0}^{T-1}f(\theta_i))$. Subsequently we have

$$\mathbb{E}[f(\bar\theta_T) - f_*] \le \frac{1}{T}\left(\frac{\beta}{\rho(1-\beta)}(f(\theta_0) - f_*) + \frac{1-\beta}{2\alpha N\rho}\|\theta_0 - \theta_*\|^2\right)$$

$$+ \frac{1-\beta}{2\alpha N\rho}M$$

Finally, when $\alpha \in (0, \frac{1-\beta}{(N+2)L}]$, we can take $\rho = 1$, use L-smooth condition again and substitute $M$, which gives

$$\mathbb{E}[f(\bar\theta_T) - f_*] \le \frac{1}{2T}\left(\frac{L\beta}{1-\beta} + \frac{1-\beta}{\alpha N}\right)\|\theta_0 - \theta_*\|^2$$

$$+ \frac{\alpha\sigma^2}{(1-\beta)} + \frac{\alpha\sigma_g^2}{(1-\beta)} + \frac{\alpha\sigma_m^2}{2N(1-\beta)} + \frac{(1+2\beta^2)\sigma_p^2}{2\alpha N(1-\beta)}$$

It's noted that when $N = 1$, we get the basic quantized SGDM without gradient accumulation, whose convex convergence result is

$$\mathbb{E}[f(\bar\theta_T) - f_*] \le \frac{1}{2T}\left(\frac{L\beta}{1-\beta} + \frac{1-\beta}{\alpha}\right)\|\theta_0 - \theta_*\|^2$$

$$+ \frac{\alpha\sigma^2}{(1-\beta)} + \frac{\alpha\sigma_g^2}{(1-\beta)} + \frac{\alpha\sigma_m^2}{2(1-\beta)} + \frac{(1+2\beta^2)\sigma_p^2}{2\alpha(1-\beta)}$$

$\square$

## B    EXPERIMENTS ON TRANSFORMERS

Tab. 4 contains the accuracy of RoBERTa-Large(Liu et al., 2019) under LPMM framework on GLUE(Wang et al., 2018a) dataset. Each column is the median performance metrics (accuracy, F1, etc.) across 5 random seeds. Similar to LPMM-SGDM, the LPMM-Adam adopt linear quantization on parameters and gradients, nonlinear quantization on momentum and square momentum. The results show that consistent 8-bit LPMM-Adam could achieve (nearly) lossless accuracy compared with full-precision reference.

Table 4: Performance of RoBERTa-Large on GLUE Tasks. LPMM(12/8/8/8) means that 12-bit parameter, 8-bit gradient, 8-bit momentum, 8-bit square momentum are used.

| Model | MNLI | QNLI | QQP | RTE | SST-2 | MRPC | CoLA | STS-B | Avg. |
|---|---|---|---|---|---|---|---|---|---|
| Full-precision | 90.2 | 94.7 | 92.2 | 86.6 | 96.4 | 90.9 | 68.0 | 92.4 | 88.9 |
| LPMM(12/8/8/8) | 90.5 | 94.6 | 91.9 | 84.8 | 96.4 | 90.7 | 67.1 | 92.3 | 88.5 |
| LPMM(8/8/8/8) | 90.4 | 94.6 | 91.9 | 85.6 | 96.4 | 90.0 | 64.6 | 92.1 | 88.2 |

## C    MEMORY SAVING AND COMPUTATIONAL OVERHEAD

In this section, we investigate memory saving and the overhead of LPMM. Tab. 5 shows the memory saving and throughput on several benchmarks. The model memory here is the peak memory, where all parameters, gradients and optimizer states stay in memory. All measurements are performed under standard benchmarks, i.e. ImageNet classification for ResNet-50 and GLUE for RoBERTa-Large, and standard training configurations for all benchmarks. Our LPMM-Adam saves up to 3.8 GB of GPU memory (about 72%) for RoBERTa-Large and saves 62% for ResNet-50, which make the training(or finetuning) of large models more accessible. Besides, LPMM would lead to a compuation overhead especially when the model is large, and the 8-bit Optimizer(Dettmers et al., 2021) has higher throughput than the baseline. This is due to Dettmers et al. (2021) use operator fusion techniques to integrate the optimizer update, quantization into one single operator as a CUDA kernel, while LPMM use CUDA only for quantization and PyTorch for other operations. This problem could be mitigated if the LPMM framwork is completely implemented in CUDA.

In Tab. 6, we investigate the largest trainable model in a single GPU with same memory under LPMM. We use a batch size of one and vary the depth and width respectively to explore the limit size. The GPU memory is consistently 11 GB for this comparison. Tab. 5 and 6 both show LPMM could attain a higher reduction rate for language tasks than vision tasks. This is because activation accounts more memory in vision tasks.

Table 5: Saved Memory and Throughput on Various Tasks.

| Method | Model | Model Mem | Saved Mem | Throughput(IPS) |
|---|---|---|---|---|
| 32-bit Momentum | ResNet-50 | 299.1 MB | 0.0 MB(0%) | 347.86 |
| 8-bit (Dettmers et al., 2021) | ResNet-50 | 224.4 MB | 74.7 MB(25%) | 348.71 |
| LPMM-Momentum(12/8/8) | ResNet-50 | 112.6 MB | 186.5 MB(62%) | 347.71 |
| 32-bit Adam | RoBERTa-Large | 5.3 GB | 0.0 GB(0%) | 186.2 |
| 8-bit (Dettmers et al., 2021) | RoBERTa-Large | 3.3 GB | 2.0 GB(38%) | 244.1 |
| LPMM-Adam(12/8/8/8) | RoBERTa-Large | 1.5 GB | 3.8 GB(72%) | 61.5 |

## D    COMPARISON OF LPMM WITH QUANTIZED TRAINING TECHNIQUES

To clarify the difference in contributions and techniques between LPMM and other work, including quantized training, communication efficient training, memory efficient training, etc., we summarize the computation bits and storage bits of existing works. First, we would like to point out that there are two types of numerical precision in a low-precision training system:

Table 6: Largest trainable(or finetunable) Model with same GPU memory.

| | Dim | FP | Maximum Value (Dettmers et al., 2021) | LPMM |
|---|---|---|---|---|
| ResNet-152 | D | 712 | 804 | 1016 |
| | W | 296 | 340 | 472 |
| RoBERTa-Large | D | 40 | 72 | 148 |
| | W | 1312 | 2048 | 3232 |

- *Storage bitwidth* characterizes the numerical precision of parameters, gradients, and momentums stored in the memory, where we use

  - parameter storage bitwidth (PSB) to refer the bitwidth of the stored parameters (a.k.a., primal/master parameters or latent weights);
  - gradient storage bitwidth (GSB) to refer the bitwidth of stored (parameter) gradients, which are fed into the optimizer;
  - momentum storage bit (MSB) to refer the bitwidth of stored momentums.

- *Computation bitwidth* characterizes the numerical precision of the computing kernels (GEMM, conv, etc.), where we use

  - parameter computation bitwidth (PCB) to refer the bitwidth of parameter inputs to forward operators;
  - gradient computation bitwidth (GCB) to refer the bitwidth of (activation) gradient inputs to backward operators;
  - momentum computation bitwidth (MCB) to refer the bitwidth of momentum in optimizer operators.

We list the computation and storage bits of existing works Tab. 7. Existing low-precision training methods (DoReFa, WAGE, FP8, INT8, GradScale) mainly consider reducing the computation bitwidth. The parameters are stored in high-precision (16 or 32) and then converted to low-precision (4 or 8) for computing forward and backward propagation. Among these methods, LPMM can achieve lowest storage bits. Meanwhile, those memory efficient quantized training methods (WAGEUBN, FXPNet) suffer from major accuracy loss, as shown in Tab. 8.

Table 7: Difference between LPMM and other low-precision training works. p=parameter, g=gradient, m=momentum. "-" stands for not considered or mentioned.

| Type | Method | Computation Bits | | | Storage Bits | | |
|---|---|---|---|---|---|---|---|
| | | p. | g. | m. | p. | g. | m. |
| Quantized Training | DoReFa-Net(Zhou et al., 2016) | 1 | 6 | - | 32 | 32 | - |
| | WAGEUBN(Yang et al., 2020) | 8 | 8/16 | 13 | 24 | **8** | 15 |
| | FP8 training(Wang et al., 2018b) | 8 | 8 | 16 | 16 | 16 | 16 |
| | INT8(Zhu et al., 2020) | 8 | 8 | - | 32 | 32 | - |
| | FXPNet(Chen et al., 2017) | 1 | 12 | 32 | **12** | 12 | 12 |
| | GradScale(Sun et al., 2020) | 4 | 4 | - | 16/32 | 16 | - |
| Communication Efficient Training | QSGD(Alistarh et al., 2017) | 32 | 32 | - | 32 | **4/8** | - |
| | signSGD(Bernstein et al., 2018) | 32 | 32 | - | 32 | **1** | - |
| Memory Efficient Training | LP-SGD(Li et al., 2017) | 32 | 32 | - | **1** | 32 | - |
| | HALP(De Sa et al., 2018) | 32 | 32 | - | 32 | 32 | - |
| | SWALP(Yang et al., 2019) | 32 | 32 | 32 | **9/32** | **8** | **8** |
| | 8-bit Optim.(Dettmers et al., 2021) | 32 | 32 | 32 | 32 | 32 | **8** |
| | LPMM(ours) | 32 | 32 | 32 | **12** | **8** | **8** |

Next, we compare LPMM with existing methods in more detail at the aspects of contributions, experiments and how stochastic rounding is used.

**Difference on Contributions**   First, we talk about the quantized training. DoReFa-Net, INT8 and GradScale mainly focus on the acceleration of the training process.  While they also can reduce the training memory footprint via low-bit kernel, the reduction is nearly independent to the model size. On the other hand, FP8 training, WAGEUBN and FXPNet focus on both training acceleration and memory footprint reduction (of the model storage). For communication efficient training, they quantize the gradient into lower bits for more efficient communication in distributed scenario. While the goals are different, both communication efficient training and LPMM use a lower GSB, which is necessary and beneficial to persistent storage of gradients.

For memory efficient training, we mainly refer to the work focusing on the quantization on post-update parameters and other values. LP-SGD only stays in theoretical analysis. HALP quantized the post-update parameters though, they store the main parameters in full-precision and in fact are intended to use low-precision computations. SWALP extended the HALP but in fact stored high-precision parameters as well.  Unlike others, 8-bit Optimizer focused on reducing the storage of optimizer states instead of primal parameters, in fact the two are proportional (even equal). Actually, WAGEUBN, FXPNet and SWALP did not report the memory saving which probably indicates they only conducted simulation experiments, and to our best knowlegde, we are the first to store low-bit parameters with <16-bit in realistic scenarios.

**Difference on Experimental Results**   Here we mainly talk about the experiments related to low-precision model memory.  As shown in Tab. 8, the performance of WAGEUBN and SWALP are lossy compared with full-precision baseline on ImageNet dataset, while FP8 training and GradScale are nearly lossless on both CIFAR and ImageNet under their mainly claimed setting. The FXPNet has lossless accuracy on CIFAR but no experiments on ImageNet.  Actually, there is an intrinsic complexity gap between the two datasets. From our observation, 8-bit primal parameters are enough for CIFAR10 to attain lossless accuracy, but 12-bit parameters are needed for ImageNet.

In other words, to our best knowledge, there are no comparable performance with full-precision baseline under the primal parameters less than 16-bit in previous work. However, LPMM attain lossless accuracy using 12-bit parameter. For other values, QSGD and 8-bit Optimizer can achieve lossless performance compared with full-precision using 4/8-bit gradient and 8-bit optimizer states, respectively.

Table 8: Comparison of experiments between LPMM and comparable previous work for ResNet on ImageNet

| Method | Model | Storage Bits | | | Acc. | FP Baseline |
|---|---|---|---|---|---|---|
| | | p. | g. | m. | | |
| WAGEUBN(Yang et al., 2020) | ResNet-18 | 24 | 8 | 15 | 67.40 | 68.70 |
| FP8 training(Wang et al., 2018b) | ResNet-18 | 16 | 16 | 16 | 66.95 | 67.43 |
| FXPNet(Chen et al., 2017) | - | 12 | 12 | 12 | - | - |
| GradScale(Sun et al., 2020) | ResNet-18 | 16/32 | 16 | - | 68.27 | 69.40 |
| SWALP(Yang et al., 2019) | ResNet-18 | 9/32 | 32 | 32 | 65.11 | 69.51 |
| LPMM(ours) | ResNet-18 | 12 | 8 | 8 | 71.0 | 71.1 |
| LPMM(ours) | ResNet-18 | 8 | 8 | 8 | 69.2 | 71.1 |
| QSGD(Alistarh et al., 2017) | ResNet-50 | 32 | 4/8 | 8 | 74.76 | 74.68 |
| 8-bit Optim.(Dettmers et al., 2021) | ResNet-50 | 32 | 32 | 8 | 77.2 | 77.1 |
| LPMM(ours) | ResNet-50 | 12 | 8 | 8 | 77.3 | 77.2 |

**How SR is used**   Then, we discuss about how stochastic rounding (Gupta et al., 2015) is used in various work.  Firstly, FXPNet and GradScale did not use stochastic rounding technique in their main contribution.  DoReFa-Net, WAGEUBN, INT8 only use stochastic rounding in the quantizer before forward and backward operations to generate a low-bit input to the low-bit kernel, which is completely different from our setting. The FP8 training, HALP, SWALP uses stochastic rounding in parameter update, same as LPMM. However, we dive deeply into why stochastic rounding is necessary from network training dynamics and parameter distribution, while they either give accuracy gap with or without stochastic rounding, or not concern about the SR technique itself. Lastly, 8-bit Optimizer and signSGD did not use stochastic rounding, and QSGD used stochastic rounding on

gradients, which we think possibly make a different roles from that on parameters since the difference on distributions. Again, QSGD also did not concern about the why SR is necessary. In short, we empirically analyze that why the unbiasedness of SR is necessary from the training dynamics and parameter distribution, which is not given in previous work.

