# OpenReview forum: "Training Neural Networks with Low-Precision Model Memory"
_ICLR.cc/2023/Conference — Submitted to ICLR 2023_

### Official Review · Reviewer_wNyQ · 2022-10-23

**Confidence:** 5
**Correctness:** 3
**Technical Novelty And Significance:** 1
**Empirical Novelty And Significance:** 2
**Recommendation:** 3

**Clarity, Quality, Novelty And Reproducibility:**

- Two major ideas of the manuscript seem to have been already introduced previously. What is new compared to those previous works?
- Improvements and novelty are incremental at best.
- Only two models (ResNet-18 and ResNet-50) are analyzed. A lot more models and dataset need to be investigated to claim the generality of the proposed techniques.

**Strength And Weaknesses:**

*Strength
- Convergence of the proposed low-precision training is analyzed.
- The impact of microbatching is investigated with details.

*Weakness
- Stochastic rounding has been introduced by lots of previous works. What is the major difference compared to those related works? For example, the paper titled "Training deep neural networks with 8-bit floating point numbers" already introduced while the first paper introducing stochastic rounding is probably "Deep Learning with Limited Numerical Precision" which is missing in the reference.
- Properties of microbatching seem to be very similar to chunk-based additions of the above reference.
- Comparisons with previous low-precision training are missing in the experiments.


**Summary Of The Paper:**

The authors propose low-precision training techniques to lower the number of bits to represent parameters, gradients, and moments. SGDM optimizer is assumed while low-precision training can improve the model accuracy by using two main principles: unbiased stochastic quantization and microbatching. Overall, memory consumption can be reduced by over 70% when ResNet-18 and ResNet-50 (on ImageNet) are trained.

**Summary Of The Review:**

- Ideas are incremental (or not new at all)
- Experiments need to be more extensive. Only two models are not enough to prove the claim.

---

> ### Author Response · Authors · 2022-11-18
> **Official Comment by Paper5887 Authors (2/2)**
>
> # Q3: Comparisons with previous low-precision training are missing in the experiments.
> We compare with the previous related work from the reported experiment results In Appendix D. We compare the main difference in Tab.7 and experimental difference in Tab.8, and we also list them below.
> In short, low-bit training methods keep their storage bitwidth high (16 or 32 bits), with the exceptions WAGEUBN & FXPNet have major accuracy loss. In contrast, LPMM can reduce the storage bitwidth while retaining the accuracy of full-precision optimizers.
>
> The table below shows the difference between LPMM and other low-precision training works. p=parameter, g=gradient, m=momentum, c=computation, s=storage. ''-'' stands for not considered or mentioned. These metrics are defined in detail in Appendix D.
>
> | Type | Method | p. c. bit | g. c. bit | m. c. bit | p. s. bit | g. s. bit | m. s. bit |
> | :-----| :---- | :---- | :-----| :---- | :---- | :---- | :-----|
> | Quantized Training | DoReFa-Net   | 1     | 6     | -     | 32    | 32    | -     |
> | Quantized Training | WAGEUBN      | 8     | 8/16  | 13    | 24    | **8** | 15    |
> | Quantized Training | FP8 training | 8     | 8     | 16    | 16    | 16    | 16    |
> | Quantized Training | INT8         | 8     | 8     | -     | 32    | 32    | -     |
> | Quantized Training | FXPNet       | 1     | 12    | 32    | **12**| 12    | 12    |
> | Quantized Training | GradScale    | 4     | 4     | -     | 16/32 | 16    | -     |
> | Communication Efficient Training | QSGD     | 32    | 32    | -     | 32    | **4/8**   | - |
> | Communication Efficient Training | signSGD  | 32    | 32    | -     | 32    | **1**   | - |
> | Memory Efficient Training | LP-SGD  | 32    | 32    | -     | **1** | 32    | - |
> | Memory Efficient Training | HALP    | 32    | 32    | -     | 32    | 32    | - |
> | Memory Efficient Training | SWALP   | 32    | 32    | 32    | **9**/32 | **8** | **8** |
> | Memory Efficient Training | 8-bit Optim. | 32    | 32    | 32     | 32 | 32 | **8** |
> | Memory Efficient Training | LPMM | 32    | 32    | 32     | **12** | **8** | **8** |
>
>
> The table below compares the experiments between LPMM and other comparable works, i.e. focus memory reduction. p=parameter, g=gradient, m=momentum, are all storage bits. The table is also shown in Appendix D.
>
>
> |  Method | Model | p. | g. | m. | Acc. | FP basline |
> | :-----| :---- | :---- | :-----| :---- | :---- | :---- |
> | WAGEUBN      | ResNet-18      | 24    | 8     | 15    | 67.40 | 68.70 |
> | FP8 training | ResNet-18      | 16    | 16    | 16    | 66.95 | 67.43 |
> | FXPNet       | -              | 12    | 12    | 12    | -     | -     |
> | GradScale    | ResNet-18      | 16/32 | 16    | -     | 68.27 | 69.40 |
> | SWALP        | ResNet-18      | 9/32  | 8     | 8     | 65.11 | 69.51 |
> | LPMM(ours)   | ResNet-18      | 12    | 8     | 8     | 71.0  | 71.1  |
> | LPMM(ours)   | ResNet-18      | 8     | 8     | 8     | 69.2  | 71.1  |
> | QSGD         | ResNet-50      | 32    | 4/8   | -     | 74.76 | 74.68 |
> | 8-bit Optim. | ResNet-50      | 32    | 32    | 8     | 77.2  | 77.1  |
> | LPMM(ours)   | ResNet-50      | 12    | 8     | 8     | 77.3  | 77.2  |
>
>
> Thanks again for your review!

---

> > ### Author Response · Authors · 2022-12-01
> > **Sincerely looking forward to the further discussions**
> >
> > Dear reviewer,
> >
> > We are wondering if our response and revision have resolved your concerns. If our response has addressed your concerns, we would highly appreciate it if you could re-evaluate our work and consider raising the score.
> >
> > If you have any additional questions or suggestions, we would be happy to have further discussions.
> >
> > Best regards,
> >
> > The Authors

---

> ### Author Response · Authors · 2022-11-18
> **Official Comment by Paper5887 Authors (1/2)**
>
> Thanks for the insightful and constructive feedback! Below, we address the comments:
>
> # Q1: Stochastic rounding has been introduced by lots of previous works. What is the major difference compared to those related works?
>
> We add a detailed discussion about how SR is applied for plentiful low-precision training work in Appendix D. We also list it below.
>
>     Firstly, FXPNet and GradScale did not use stochastic rounding technique in their main contribution. DoReFa-Net, WAGEUBN, INT8 only use stochastic rounding in the quantizer before forward and backward operations to generate a low-bit input to the low-bit kernel, which is completely different from our setting. The FP8 training, HALP, SWALP uses stochastic rounding in parameter update, same as LPMM. However, we dive deeply into why stochastic rounding is necessary from network training dynamics and parameter distribution, while they either give accuracy gap with or without stochastic rounding, or not concern about the SR technique itself. Lastly, 8-bit Optimizer and signSGD did not use stochastic rounding, and QSGD used stochastic rounding on gradients, which we think possibly make a different roles from that on parameters since the difference on distributions. Again, QSGD also did not concern about the why SR is necessary. In short, we empirically analyze that why the unbiasedness of SR is necessary from the training dynamics and parameter distribution, which is not given in previous work.
>
> As we discussed in Appendix D, only some of them use stochastic rounding in parameter update. However, all of them do not discuss about why SR is necessary for network training. The work [1] only discuss about how SR relates to mitigate the accumulation error, while we analyze SR from the perspective of networking. Specifically, we analyze SR by network training dynamics(Fig.2) and parameter distribution (Fig.3).
>
> [1] https://arxiv.org/pdf/1812.08011.pdf
>
> # Q2: Properties of microbatching seem to be very similar to chunk-based additions of the above reference.
>
> We agree with the reviewer that [1] also extensively discussed underflowing and inspire our work.
> However, the solution for swamping issue in [1], i.e. the chunk-based accumulation is different from the our taken microbatching. The chunk-based accumulation mainly discusses the underflowing in dot products (it does discuss parameter updates though). Instead, the goal of microbatching is to accumulate tiny gradient updates over time, without keeping a high-precision version of the parameter.
>
> Additionally, we argue that we study the problem of parameter update more specifically and thoroughly than previous works.

---

### Official Review · Reviewer_Dafx · 2022-10-25

**Confidence:** 3
**Correctness:** 3
**Technical Novelty And Significance:** 2
**Empirical Novelty And Significance:** 3
**Recommendation:** 6

**Clarity, Quality, Novelty And Reproducibility:**

The paper is well structured, clear, and easy to follow.
While the techniques used to quantize the model parameters and address underflow are known, there is novelty in how they are jointly applied to this case study (CNNs) and how convergence is theoretically demonstrated.
Implementation is explained in details so results should be reproducible.



**Strength And Weaknesses:**

Strengths:
- combination of stochastic rounding (SR) and microbatching achieves remarkable model memory compression with no (or limited) accuracy degradation
- good theoretical support on LPMM-SGDM convergence

Weaknesses:
- the paper heavily relies on known techniques (SR and microbatching) which are applied as-is to the particular case study
- the main insight is claimed to be identification of underflow as the core issue to be solved. However, the impact of delays in the parameters update (or lack thereof) is a well known quantization-related issue, both in the case of having access or not to a full precision copy of the quantized parameters (see for example the already cited [1], which discusses the conceptually-similar swamping issue as well as SR, or [2], which discusses mitigation strategies for updates delayed by quantization). So, it's hard to accept the claim that it has been "discovered" in this particular scenario
- demonstration is limited to two CNN models so it's unclear if findings may generalize (e.g., is this compression level suitable for other models?)
- what is the runtime (or training time overhead) of this algorithm? how practical is it to apply this training technique beyond the two examples shown?
- there is a mention of a training algorithm implementation at the level of CUDA kernel in PyTorch but no details about it

Other corrections:
- section 3.2: part of sentence missing: "...is far from the Extremely, there could be"
- algorithm 1: should the quantization in line 5 be carried out _after_ all the micro-batch gradients have been accumulated? Doesn't applying the quantization at every step defeats the purpose of accumulation if gradients are individually small?

[1] N. Wang et al. "Training Deep Neural Networks with 8-bit Floating Point Numbers" (section 3.2)
[2] M. Nagel et al. “Overcoming Oscillations in QAT”

**Summary Of The Paper:**

This paper addresses the issue of large model memory as a bottleneck to model scaling. It proposes a framework to compress model parameters, gradient accumulations, and momentum, during training. Achieves iso-accuracy on ResNet/Imagenet models down to 12/8/8 bits, respectively, and minimal degradation down to 8/8/8 bits. Underflow is addressed by combining stochastic rounding with microbatching (accumulation of subsequent gradients). Convergence is mathematically guaranteed in the presence of stochastic quantization.

**Summary Of The Review:**

The paper applies known techniques to successfully compress model memory (i.e., model parameters, gradient accumulations, and momentum) and achieve strong compression and iso-accuracy in two CNN models. I would like to see more details about the algorithm implementation and its runtime (possibly an additional appendix).

---

> ### Author Response · Authors · 2022-11-18
> **Official Comment by Paper5887 Authors (3/3)**
>
> # Q5: Implementation at the level of CUDA kernel in PyTorch
> In fact, the CUDA kernel we used is only about the quantization and de-quantizaton operation.
> Since the computation bit of LPMM is 32-bit, as newly listed in Appendix D, the forward and backward computation is intrinsically the same as full-precision counterpart. Under the support of autograd engine in PyTorch, we found the only thing needed using the CUDA kernel is the quantizer. We think the implementation of the quantizer is not technically difficult. For example, for linear quantization, we just scale every tensor with a known scaling factor, and do stochastic rounding to a integer.
>
> # Q6: Doesn't applying the quantization at every step defeats the purpose of accumulation if gradients are individually small?
> Great question! We exactly apply the quantization on the accumulated gradients at every step. Firstly, to achieve complete memory efficient training, we must quantize the accumulated gradient at every step. Otherwise, the accumulation would incur extra persistent memory storage of gradients, which is exactly equal to the model size. And If the accumulation is not used, this can be avoided by adding gradient to momentum or parameters as soon as one backward operator finishes.
> Secondly, the accumulation of gradients and the update to parameters are two things, and we found the underflow would occur in the last situation, hence we use the former trying to mitigate the underflow. Even if the gradients are individually small, the accumulation would not collapse since the scaling factor(or quantization interval) for gradients is different from that for parameters. The scaling factor of gradients would match the magnitude of gradients which ensure the accumulation works well.
>
> Thanks again for your review!

---

> > ### Author Response · Authors · 2022-12-01
> > **Sincerely looking forward to the further discussions**
> >
> > Dear reviewer,
> >
> > We are wondering if our response and revision have resolved your concerns. If our response has addressed your concerns, we would highly appreciate it if you could re-evaluate our work and consider raising the score.
> >
> > If you have any additional questions or suggestions, we would be happy to have further discussions.
> >
> > Best regards,
> >
> > The Authors

---

> > > ### Comment · Reviewer_Dafx · 2022-12-02
> > > **update**
> > >
> > > The authors did a thorough job at addressing my main concerns, including expanding the experimental section, discussing runtime of LPMM, and comparing with other works in the literature. I still see the work as incremental but in my opinion it clears the bar for novelty as it applies known techniques in a new (or at least much less investigated) context. Overall, I still lean towards acceptance.

---

> ### Author Response · Authors · 2022-11-18
> **Official Comment by Paper5887 Authors (2/3)**
>
> # Q3: Experiment is limited to two CNN model and whether compression level is suitable
>
> We add the new experiment about Adam and RoBERTa-Large finetuning task on GLUE dataset in Appendix B. We also list the results below. The results show that even consistent 8-bit LPMM-Adam (with 8-bit parameter, gradient, momentum, square momentum) can achieve nearly lossless performace compared with full-precision counterpart. In particular, the compression level is tighter than image classification task.
>
> |Model  | MNLI | QNLI | QQP | RTE | SST-2 | MRPC | CoLA | STS-B | Avg. |
> | :-----| :---- | :---- | :-----| :---- | :---- | :---- | :-----| :-----| :-----|
> |Full-precision  | 90.2  | 94.7 | 92.2 | 86.6 |	96.4 | 90.9 | 68.0 | 92.4  | 88.9 |
> LPMM-Adam(12/8/8/8)  | 90.5 | 94.6 | 91.9 | 84.8 | 96.4 | 90.7 | 67.1 | 92.3 | 88.5 |
> LPMM-Adam(8/8/8/8)   | 90.4 | 94.6 | 91.9 | 85.6 | 96.4 | 90.0 | 64.6 | 92.1 | 88.2 |
>
> # Q4: Runtime overhead of LPMM
>
> We complement the experiments about memory saving and runtime overhead in Appendix C. The result is also listed below. The results show LPMM would incur run time overhead, especially when the model is large, and the 8-bit Optim. has higher throughput than the baseline.
> This is due to 8-bit Optim. use operator fusion techniques to integrate the optimizer update, quantization into one single operator as a CUDA kernel, while LPMM use CUDA only for quantization and PyTorch for other operations.
> This problem could be mitigated if the LPMM framwork is completely implemented in CUDA.
>
> |Method  | Model | Model Mem | Saved Mem | Throughput(IPS) |
> | :-----| :---- | :---- | :-----| :---- |
> | 32-bit Momentum   | ResNet-50   | 299.1 MB | 0.0 MB(0\%)    | 347.86  |
> |8-bit Optim. (Dettmers et. al, 2021) |   ResNet-50 |224.4 MB | 74.7 MB(25\%) | 348.71 |
> |LPMM-Momentum(12/8/8)   | ResNet-50   | 112.6 MB  |186.5 MB(62\%) | 347.71  |
> |32-bit Adam          |   RoBERTa-Large  |5.3 GB | 0.0 GB(0\%)   |186.2    |
> 8-bit Optim. (Dettmers et. al, 2021) |   RoBERTa-Large   | 3.3 GB | 2.0 GB(38\%)  | 244.1    |
> |LPMM-Adam(12/8/8/8)      |   RoBERTa-Large   |1.5 GB | 3.8 GB(72\%)   | 61.5     |

---

> ### Author Response · Authors · 2022-11-18
> **Official Comment by Paper5887 Authors (1/3)**
>
> Thanks for the careful view and constructive feedback! Below, we address the comments:
>
> # Q1: Heavily relies on known techniques
> The main problem we address is the *design and analysis of low-bit optimizers*, which differs from most existing low-bit training methods. Designing a low-bit optimizer is highly non-trivial. Most existing low-bit training works maintain a high-precision master weight (a.k.a., primal / latent weight) to accumulate gradient updates. However, the master weight is not affordable in our scenrio. This raises an unique challenge: how to accumulate tiny gradient updates over time, without keeping a high-precision version of the parameter?
>
> To this end, we identify this unique challenge and propose solutions accordingly. The main novelty is not proposing the techniques (microbatching, stochastic rounding), but finding suitable techniques to solve our unique challenge. Furthermore, we provide a rigorous theoretical analysis of our approach, which not only guarantee the convergence, but not justify the effectiveness of our utilized techniques. To the best of our knowledge, LPMM is the first low-bit optimizer with guaranteed convergence with all parameters, momentum, and gradients quantized.
>
> # Q2: Identifying underflow as the core issue is not novel.
> We agree with the reviewer that [1, 2] also extensively discussed underflowing and inspire our work.
>
> However, underflowing happens in different parts of the computation:
> - [1] mainly discusses the underflowing in dot products (it does discuss parameter updates though);
> - [2] discusses the osciallation of the quantized weight. It still keeps a full-precision "master" weight.
>
> We argue that we study the problem of parameter update more specifically and thoroughly than previous works.
>
> For our parameter update problem, underflowing is not the only possible challenge. It could also be
> - the change of numerical range of parameters during training; or
> - the error (or variance) of repeatly quantizing parameters in each iteration.
>
> So we feel that there are still information gain of pointing out that the core issue is indeed underflowing, not the above listed ones. We will revise our paper to more properly acknowledge previous works' contributions. Thanks for the suggestion!
>
> [1] https://arxiv.org/pdf/1812.08011.pdf
>
> [2] https://arxiv.org/pdf/2203.11086.pdf

---

### Official Review · Reviewer_UjMp · 2022-10-27

**Confidence:** 4
**Correctness:** 4
**Technical Novelty And Significance:** 2
**Empirical Novelty And Significance:** 3
**Recommendation:** 5

**Clarity, Quality, Novelty And Reproducibility:**

Clarity: the paper is well-written and clear.

Originality: the work mostly applies well-known quantization techniques

Reproducibility: there should be sufficient detail to reproduce the results.

**Strength And Weaknesses:**

Strengths:
 - Appears to be the first paper to quantize params, gradients, and optimizer state jointly

Weaknesses
 - Experiment section is weak. ImageNet is no longer a SOTA benchmark and ResNet is an 8-year old model that is known to be easy to quantize and train. Experiments focus only on SGD with momentum, but currently ADAM is necessary for Transformers.
 - The techniques applied are well known. For example, the Deepmind Gopher paper [1] used stochastic rounding during parameter updates. Microbatching is already implemented as a toggle option in HuggingFace and Megatron Transformer frameworks. These techniques are not really used in a novel or insightful way.
 - The technique seems difficult to implement and may incur run time overhead. The authors had to rely on non-linear quantization for the momentum, which is quite complex. Unbiased stochastic quantization requires generating random numbers and comparing them to a high-precision rounding probability for each element. These two issues may cause run time overhead on GPU and present difficulties for dedicated hardware. I would like the authors to discuss the actual memory saved on GPU (not just bitwidth) and whether LPMM reduces wall-clock time for training.

[1] https://arxiv.org/pdf/2112.11446.pdf

**Summary Of The Paper:**

The authors propose low-precision model memory (LPMM), which quantizes a model's parameters, gradients, and optimizer state to low precision. Prior works have focused only on params/gradients or optimizer state separately. The main challenge is that params, gradients, and momentum may have very different dynamic range, leading to underflow (typically of the gradient updates). LPMM addresses this via unbiased stochastic rounding and increasing the batch size to increase the magnitude of the updates. The paper uses the term "microbatching" but this is just accumulating multiple batches of gradients before computing a weight update (some frameworks call this gradient accumulation).

The authors experiment with ResNet-18 and ResNet-50 on ImageNet, showing that LPMM with 8-bits for params, grads, and momentum can get within 2% of the full precision accuracy. If the params are kept in 12 bits then they can match full precision.

EDIT: raised score from 3 to 5 after rebuttal.

**Summary Of The Review:**

The paper applies stochastic quantization and microbatching to quantize parameters, gradients, and optimizer state. These techniques are well known in DNN quantization and thus the paper is a bit lacking in novelty. The experiment section is weak with only results for ResNets on ImageNet. Lastly, the techniques may be difficult to implement and/or incur run time overhead, which is currently not discussed.

---

> ### Author Response · Authors · 2022-11-18
> **Official Comment by Paper5887 Authors (2/2)**
>
> # Q3: The actual memory saved on GPU and runtime overhead.
> We add experiments with memory saving and computation overhead in Appendix C. The Tab.5 show that LPMM could reduce over 60\% and 70\% of the model memory on GPU compared with the full-precision SGDM and Adam, respectively. Additionally, we also investigate the largest trainable model with same GPU memory. By the Tab.6, LPMM exactly makes the training(or finetuning) of large models more accessible. We also list the tables below.
>
> This table shows the memory saving and throughput on various standard tasks.
>
> |Method  | Model | Model Mem | Saved Mem | Throughput(IPS) |
> | :-----| :---- | :---- | :-----| :---- |
> | 32-bit Momentum   | ResNet-50   | 299.1 MB | 0.0 MB(0\%)    | 347.86  |
> |8-bit Optim. (Dettmers et. al, 2021) |   ResNet-50 |224.4 MB | 74.7 MB(25\%) | 348.71 |
> |LPMM-Momentum(12/8/8)   | ResNet-50   | 112.6 MB  |186.5 MB(62\%) | 347.71  |
> |32-bit Adam          |   RoBERTa-Large  |5.3 GB | 0.0 GB(0\%)   |186.2    |
> 8-bit Optim. (Dettmers et. al, 2021) |   RoBERTa-Large   | 3.3 GB | 2.0 GB(38\%)  | 244.1    |
> |LPMM-Adam(12/8/8/8)      |   RoBERTa-Large   |1.5 GB | 3.8 GB(72\%)   | 61.5     |
>
> This table shows the largest trainable(or finetunable) model with same GPU memory.
>
> |   | Dim | FP | (Dettmers et. al, 2021) | LPMM |
> | :---- | :---- | :---- | :-----| :---- |
> |ResNet-152  | D     | 712   | 804   |1016  |
> |ResNet-152 | W   | 296   | 340   |472 |
> |RoBERTa-Large| D   | 40    | 72    |148 |
> |RoBERTa-Large | W   | 1312  | 2048  | 3232  |
>
> On the other hand, LPMM would incur run time overhead, especially when the model is large, and the 8-bit Optim. has higher throughput than the baseline.
> This is due to 8-bit Optim. use operator fusion techniques to integrate the optimizer update, quantization into one single operator as a CUDA kernel, while LPMM use CUDA only for quantization and PyTorch for other operations.
> This problem could be mitigated if the LPMM framwork is completely implemented in CUDA.
>
> Thanks again for your review!

---

> > ### Comment · Reviewer_UjMp · 2022-11-28
> > **Thanks for conducting additional experiments**
> >
> > The additional data in the rebuttal is great and adds a much stronger case to the paper. The ability to train larger models especially makes the technique valuable even in cases where the run time reduction is not significant. I maintain my stance that the non-linear momentum quantization seems unwieldy and adds overhead, and that the paper is not too novel.

---

> > > ### Author Response · Authors · 2022-12-01
> > > **Thank you for raising the score!**
> > >
> > > Thank you so much for raising the score. We are happy to see that we could resolve your concern.

---

> ### Author Response · Authors · 2022-11-18
> **Official Comment by Paper5887 Authors (1/2)**
>
>
> Thank you for your detailed and insightful review! Below, we add the comments.
>
> # Q1: Experiment section is weak. Adam is necessary for Transformers.
> We add the experiments of Adam optimizer on RoBERTa-Large model under GLUE finetuning task in Appendix B.
> The results show that LPMM can extend successfully for Adam and Transformers, and even consistent 8-bit LPMM-Adam (i.e. with 8-bit parameter, 8-bit gradient, momentum, square momentum) could achieve comparable performance compared with the full-precision counterpart. The results are also listed below.
> (12/8/8/8) means 12-bit parameter, 8-bit gradient, 8-bit momentum, 8-bit square momentum are used.
>
> |Model  | MNLI | QNLI | QQP | RTE | SST-2 | MRPC | CoLA | STS-B | Avg. |
> | :-----| :---- | :---- | :-----| :---- | :---- | :---- | :-----| :-----| :-----|
> |Full-precision  | 90.2  | 94.7 | 92.2 | 86.6 |	96.4 | 90.9 | 68.0 | 92.4  | 88.9 |
> LPMM-Adam(12/8/8/8)  | 90.5 | 94.6 | 91.9 | 84.8 | 96.4 | 90.7 | 67.1 | 92.3 | 88.5 |
> LPMM-Adam(8/8/8/8)   | 90.4 | 94.6 | 91.9 | 85.6 | 96.4 | 90.0 | 64.6 | 92.1 | 88.2 |
>
> # Q2: The techniques applied are not used in a novel or insightful way.
>
> The main problem we address is the *design and analysis of low-bit optimizers*, which differs from most existing low-bit training methods. Designing a low-bit optimizer is highly non-trivial. Most existing low-bit training works maintain a high-precision master weight (a.k.a., primal / latent weight) to accumulate gradient updates. However, the master weight is not affordable in our scenrio. This raises an unique challenge: how to accumulate tiny gradient updates over time, without keeping a high-precision version of the parameter?
>
> To this end, we identify this unique challenge and propose solutions accordingly. The main novelty is not proposing the techniques (microbatching, stochastic rounding), but finding suitable techniques to solve our unique challenge. Furthermore, we provide a rigorous theoretical analysis of our approach, which not only guarantee the convergence, but not justify the effectiveness of our utilized techniques. To the best of our knowledge, LPMM is the first low-bit optimizer with guaranteed convergence with all parameters, momentum, and gradients quantized.

---

### Official Review · Reviewer_wG9u · 2022-10-27

**Confidence:** 5
**Clarity, Quality, Novelty And Reproducibility:** Please refer to strength and weaknesses
**Correctness:** 3
**Technical Novelty And Significance:** 1
**Empirical Novelty And Significance:** 1
**Recommendation:** 3

**Strength And Weaknesses:**

Strengths

1. This paper proposes the low-bit version SGD optimizer, which is a good approach towards low-bit training on the device that support low precision computation.

2. The batch size experiments in figure 3 is very interesting. The author also provide convergence analysis on the training algorithm.

3. The paper organization is clear, which makes it easy to read.


Weaknesses

1. There are many existing work on all low-bit training. And the author didn’t compare the proposed method to them. Such as WAGEUBN (Training High-Performance and Large-Scale Deep Neural Networks with Full 8-bit Integers), DoReFa-Net (Dorefa-net: Training low bitwidth convolutional neural networks with low bitwidth gradients), FXPNet (Fxpnet: Training a deep convolutional neural network in fixed-point representation), GradScale (Ultra-Low Precision 4-bit Training of Deep Neural Networks) and (Towards Unified INT8 Training for Convolutional Neural Network). They are very important research on all low-bit training and has been published in top conference/journals. Although some of them are cited in related works, but the author didn’t discuss nor compare with them.

2. Because there are many similar works, then the novelty of this work is not that significant. And I don’t see the author discuss about the problems of existing works, which makes the motivation of this work relatively weak.

3. One important part is missing, which is how batch-norm layers are used. During training, batch-norm is keeping the training, but it will change the low-bit number into fp number. Could  the author discuss this problem?



**Summary Of The Paper:**

This paper proposes low-precision model memory (LPMM) as a quantization training concept. In LPMM, all parameters, including weights, activation, gradient, and momentums are quantized to low-bit, which achieves theoretical low-bit training that saves significant memory footprints on the hardware.

**Summary Of The Review:**

To sum up, the clarity and quality of this paper need to be improved. The author of the paper did some interesting works on model quantization but fails to demonstrate them with thorough discussion and experiments. Please refer to strengths and weaknesses for more information.

I think this paper needs major revise, both on the technical contribution and experiments.

---

> ### Author Response · Authors · 2022-11-18
> **Official Comment by Paper5887 Authors (2/2)**
>
> # Q2: The novelty of this work is not that significant.
>
> As argued above, the main problem we address is the *design and analysis of low-bit optimizers*, which differs from most existing low-bit training methods. Designing a low-bit optimizer is highly non-trivial. Most existing low-bit training works maintain a high-precision master weight (a.k.a., primal / latent weight) to accumulate gradient updates. However, the master weight is not affordable in our scenrio. This raises an unique challenge: how to accumulate tiny gradient updates over time, without keeping a high-precision version of the parameter?
>
> To this end, we identify this unique challenge and propose solutions accordingly. The main novelty is not proposing the techniques (microbatching, stochastic rounding), but finding suitable techniques to solve our unique challenge. Furthermore, we provide a rigorous theoretical analysis of our approach, which not only guarantee the convergence, but not justify the effectiveness of our utilized techniques. To the best of our knowledge, LPMM is the first low-bit optimizer with guaranteed convergence with all parameters, momentum, and gradients quantized.
>
> # Q3: How batch-norm layers are used
> Actually, from the Tab.7 in the Appendix D (or the first table in Q1), we learn that the computation of LPMM is just full-precision computation, hence the forward and backward operation is the same as full-precision counterpart. For training stability, we do not choose to quantize the parameters of BN after each update. Note that this trade-off is different from the problem of BN in quantized training. In quantized training
> (or low-precision computation training), the computation in BN is complicated and delicate, and BN operator is slower, which become a significant problem. However, the overhead by not quantizing BN in our problem is only proportional to the number of parameters in BN, which is much less than the parameters in Conv and Fully-connected Layer.
>
> Thanks again for your review!

---

> > ### Author Response · Authors · 2022-12-01
> > **Sincerely looking forward to the further discussions**
> >
> > Dear reviewer,
> >
> > We are wondering if our response and revision have resolved your concerns. If our response has addressed your concerns, we would highly appreciate it if you could re-evaluate our work and consider raising the score.
> >
> > If you have any additional questions or suggestions, we would be happy to have further discussions.
> >
> > Best regards,
> >
> > The Authors

---

> ### Author Response · Authors · 2022-11-18
> **Official Comment by Paper5887 Authors (1/2)**
>
> Thanks for the insightful and constructive feedback! Below, we address the comments:
>
> # Q1: Many low-bit training work are not discussed.
> We suspect there might be some misunderstanding: the low-bit training methods mentioned by the reviewer mostly focus on reducing the *computation bitwidth*, i.e., the numerical precision of the computation kernels (GEMM, conv, etc.); but we focus on reducing the *storage bitwidth*, i.e., the numerical precision of the stored parameters, gradients, and momentums. Reducing computation bitwidth involves approximating the *forward and backward* propagation, while reducing storage bitwidth involving the design and analysis of the *optimizer*. Therefore, the efforts for reducing the computation and storage bitwidth are complementary rather than competing.
>
> We add a detailed comparison and discussion about a variety of low-bit training works in the Appendix D. The results are also listed below. In short, low-bit training methods keep their storage bitwidth high (16 or 32 bits), with the exceptions WAGEUBN & FXPNet have major accuracy loss. In contrast, LPMM can reduce the storage bitwidth while retaining the accuracy of full-precision optimizers.
>
> The table below shows the difference between LPMM and other low-precision training works. p=parameter, g=gradient, m=momentum, c=computation, s=storage. ''-'' stands for not considered or mentioned. These metrics are defined in detail in Appendix D.
>
> | Type | Method | p. c. bit | g. c. bit | m. c. bit | p. s. bit | g. s. bit | m. s. bit |
> | :-----| :---- | :---- | :-----| :---- | :---- | :---- | :-----|
> | Quantized Training | DoReFa-Net   | 1     | 6     | -     | 32    | 32    | -     |
> | Quantized Training | WAGEUBN      | 8     | 8/16  | 13    | 24    | **8** | 15    |
> | Quantized Training | FP8 training | 8     | 8     | 16    | 16    | 16    | 16    |
> | Quantized Training | INT8         | 8     | 8     | -     | 32    | 32    | -     |
> | Quantized Training | FXPNet       | 1     | 12    | 32    | **12**| 12    | 12    |
> | Quantized Training | GradScale    | 4     | 4     | -     | 16/32 | 16    | -     |
> | Communication Efficient Training | QSGD     | 32    | 32    | -     | 32    | **4/8**   | - |
> | Communication Efficient Training | signSGD  | 32    | 32    | -     | 32    | **1**   | - |
> | Memory Efficient Training | LP-SGD  | 32    | 32    | -     | **1** | 32    | - |
> | Memory Efficient Training | HALP    | 32    | 32    | -     | 32    | 32    | - |
> | Memory Efficient Training | SWALP   | 32    | 32    | 32    | **9**/32 | **8** | **8** |
> | Memory Efficient Training | 8-bit Optim. | 32    | 32    | 32     | 32 | 32 | **8** |
> | Memory Efficient Training | LPMM | 32    | 32    | 32     | **12** | **8** | **8** |
>
>
> The table below compares the experiments between LPMM and other comparable works, i.e. focus memory reduction. p=parameter, g=gradient, m=momentum, are all storage bits. The table is also shown in Appendix D.
>
>
> |  Method | Model | p. | g. | m. | Acc. | FP basline |
> | :-----| :---- | :---- | :-----| :---- | :---- | :---- |
> | WAGEUBN      | ResNet-18      | 24    | 8     | 15    | 67.40 | 68.70 |
> | FP8 training | ResNet-18      | 16    | 16    | 16    | 66.95 | 67.43 |
> | FXPNet       | -              | 12    | 12    | 12    | -     | -     |
> | GradScale    | ResNet-18      | 16/32 | 16    | -     | 68.27 | 69.40 |
> | SWALP        | ResNet-18      | 9/32  | 8     | 8     | 65.11 | 69.51 |
> | LPMM(ours)   | ResNet-18      | 12    | 8     | 8     | 71.0  | 71.1  |
> | LPMM(ours)   | ResNet-18      | 8     | 8     | 8     | 69.2  | 71.1  |
> | QSGD         | ResNet-50      | 32    | 4/8   | -     | 74.76 | 74.68 |
> | 8-bit Optim. | ResNet-50      | 32    | 32    | 8     | 77.2  | 77.1  |
> | LPMM(ours)   | ResNet-50      | 12    | 8     | 8     | 77.3  | 77.2  |

---

### Decision · Program_Chairs · 2023-01-20

**Decision:**

Reject

**Justification For Why Not Higher Score:**

While one reviewer (`Dafx`) evaluated the paper as marginally above the acceptance threshold, all other reviewers are recommending rejection. Two of the reviewers recommending rejection are highly confident about the recommendation.

To improve, the paper requires major revision and requires another round of full review. Thus AC recommends rejection following the suggestion by the reviewers.


**Justification For Why Not Lower Score:**

N/A

**Metareview: Summary, Strengths And Weaknesses:**

The paper proposes a training quantization method called low-precision model memory (LPMM).  In LPMM, model parameter / activation / gradient and optimizer states are all quantized to low-bit which could provide significant memory footprint saving on the hardware.

Main challenge is to deal with various dynamic ranges of variables.

Authors show that in ResNet trained on ImageNet, LPMM with 8-bits could get 2% of the full precision accuracy. When parameters are kept in 12 bits authors show full precision could be achieved.

Strength
- Considered problem is important problem in model quantization
- First paper to quantize parameter, gradients and optimizer state jointly, where previous work focused on them separately.
- Good theoretical support on LPMM-SGDM convergence

Weakness
- While there are other existing low-bit training works, the paper does not compare to them. Reviewer wG9u pointed out a few examples of low-bit training methods, some of which are cited but not compared.
- Related novelty of the work maybe limited and lack of comparison to existing method makes motivation of this work to be weak
- Details on how to do batch-norm is missing
- Main claim of benefit of the method is not sufficiently supported
- Proposed technical solution: stochastic rounding and micro batching is already widely used.
- Complicated to implement (non-linear quantization) and issues with run time overhead.

Overall Clarity and Quality needs to be improved